# Therapeutic Targets for Bone and Soft-Tissue Sarcomas

**DOI:** 10.3390/ijms20010170

**Published:** 2019-01-04

**Authors:** Shinji Miwa, Norio Yamamoto, Katsuhiro Hayashi, Akihiko Takeuchi, Kentaro Igarashi, Hiroyuki Tsuchiya

**Affiliations:** Department of Orthopaedic Surgery, Kanazawa University School of Medicine, Kanazawa 920-8640, Japan; norinori@med.kanazawa-u.ac.jp (N.Y.); khayashi830@gmail.com (K.H.); a_take@med.kanazawa-u.ac.jp (A.T.); kenken99004@yahoo.co.jp (K.I.); tsuchi@med.kanazawa-u.ac.jp (H.T.)

**Keywords:** sarcoma, immunotherapy, chemotherapy

## Abstract

Due to the rarity and heterogeneity of bone and soft-tissue sarcomas, investigation into molecular targets and new treatments has been particularly challenging. Although intensive chemotherapy and establishment of surgical procedures have improved the outcomes of patients with sarcoma, the curative rate of recurrent and metastatic sarcomas is still not satisfactory. Recent basic science research has revealed some of the mechanisms of progression and metastasis of malignancies including proliferation, apoptosis, angiogenesis, tumor microenvironment, migration, invasion, and regulation of antitumor immune systems. Based on these basic studies, new anticancer drugs, including pazopanib, trabectedin, eribulin, and immune checkpoint inhibitors have been developed and the efficacies and safety of the new drugs have been assessed by clinical trials. This review summarizes new molecular therapeutic targets and advances in the treatment for bone and soft tissue sarcomas.

## 1. Introduction

Sarcomas are rare types of malignancies that account for approximately 1% of the total malignancies reported [1]. Despite the rarity, sarcomas are heterogenous malignancies of mesenchymal origin and are classified into more than 100 distinct subtypes [2]. Before introduction of chemotherapy, long-term survival was observed in only 20–40% of patients with bone sarcomas and in only 35% of patients with soft tissue sarcomas [3]. Since the 1970s, chemotherapy significantly improved the outcomes of sarcoma, and five-year survival of 60–80% have been reported for patients receiving chemotherapy and with surgical resection [4,5,6,7,8]. Although introduction of chemotherapy significantly improves the outcomes of sarcomas, only surgical resection can offer a cure for the sarcoma. Metastatic lesions can be detected in approximately 10% of patients with sarcomas at the time of diagnosis. Furthermore, 25% of patients with sarcomas develop metastatic disease after curative treatment for the primary tumor [9]. Novel treatment approaches are urgently needed for significant development of sarcoma treatment. However, development and investigation for new treatment approaches in patients with sarcoma has several limitations, including small sample size, variety of tumor subtypes, and heterogeneity in patient population.

In current standard chemotherapy, anticancer agents such as doxorubicin, ifosfamide, dacarbazine, gemcitabine, and docetaxel have been used for the treatment of sarcomas. Since the first study on doxorubicin for malignancy was reported in 1973 [10], doxorubicin remains a pivotal agent, and has been generally accepted as standard first-line drug in the therapy for advanced soft-tissue sarcomas. Clinical studies for several agents including ifosfamide and dacarbazine did not demonstrate significant advantage over doxorubicin [11,12]. Although doxorubicin-based regimens have been commonly accepted for the treatment of soft-tissue sarcomas, there has been no definitive answer to the question of monotherapy or combination therapy. A recent phase III trial comparing doxorubicin and ifosfamide with doxorubicin monotherapy was reported in 2014 (EORTC62012) [13]. The study demonstrated that doxorubicin and ifosfamide improved the response rate and progression-free survival in patients, but did not show any significant overall survival advantage compared to doxorubicin monotherapy.

Recently, new treatments for bone and soft-tissue sarcomas, including molecular targeting agents, immune checkpoint inhibitors, and adoptive T-cell therapy have been developed, and their safety and efficacies have been investigated in clinical studies. In this review article, we discuss the recent advancements in the treatment for bone and soft-tissue sarcoma.

## 2. Clinical Trials for New Anticancer Agents

### 2.1. Current and Emerging Targeting therapies for Bone and Soft Tissue Tumors

In recent years, there have been significant advances in the understanding of pathogenesis and progression of bone and soft tissue tumors. Furthermore, based on the basic studies, new targeting therapies have been developed. Since tumor growth and dissemination of malignancies require angiogenesis, controlling angiogenesis is thought to be essential for the management of sarcomas. Association of increased expression of vascular endothelial growth factor (VEGF) in soft-tissue sarcoma with higher malignancy grade and worse oncological outcomes has been well-reported [14,15]. In addition to VEGF, platelet-derived growth factor (PDGF) is also known to be involved in angiogenesis in soft-tissue sarcoma [16]. Therefore, receptors of VEGF and PDGF have been considered as target kinases for the treatment of soft-tissue sarcoma.

Pazopanib is an oral multitarget inhibitor that has been known to block vascular endothelial growth factor receptors, VEGFR-1, VEGF-R2, and VEGFR-3; and platelet-derived growth factor receptor (PDGFR) and c-Kit, and exhibits antiangiogenic and antitumor activity [17]. The efficacy and safety of pazopanib were evaluated in several clinical trials (Table 1). In a phase I study with pazopanib, the tolerability and recommended dose of pazopanib were assessed in 63 patients [18]. A phase II study with pazopanib in patients with sarcoma was reported in 2009 by European Organization for Research and Treatment of Cancer-Soft Tissue and Bone Sarcoma Group (EORTC-STBSG) (EORTC study 60243) [19]. In this study, 142 patients with intermediate- or high-grade advanced soft-tissue sarcoma underwent a daily administration of 800 mg pazopanib. Progression-free rates at 12 weeks were, 44% (18/41) in patients with leiomyosarcoma, 49% (18/37) in patients with synovial sarcoma, and 39% (16/41) in patients with other sarcomas. In 2012, EORTC group reported a randomized, double-blind, placebo-controlled phase III study (PALETTE study) with pazopanib in patients with soft-tissue sarcoma excluding liposarcoma. In the study, 369 patients with metastatic sarcoma were treated with pazopanib (800 mg) once daily or with placebo [20]. Progression-free survival rates in patients with pazopanib and placebo group were 4.6 months and 1.6 months, respectively (*p* < 0.0001). Overall survival rates of patients treated with pazopanib and placebo group were 12.5 months and 10.7 months, respectively (*p* = 0.25). Although the PALETTE study demonstrated significant improvement in progression-free survival in patients with soft-tissue sarcoma, the efficacy of pazopanib in liposarcomas was unclear because of the exclusion of patients with liposarcoma from the treated group. On the other hand, Samuels et al. reported a phase II study with pazopanib for 41 patients with liposarcoma [21]. In this study, median progression-free survival and overall survival periods were 4.4 months and 12.6 months, respectively. This study reported 3 deaths due to possible complications of the treatment. This study demonstrated that pazopanib can be considered as one of the treatment options in patients with advanced liposarcoma. In a retrospective study investigating the efficacy and safety of pazopanib in 156 patients with recurrent or metastatic soft-tissue sarcoma, median progression-free survival and overall survival periods were 15 weeks and 11 months, respectively [22]. In this study, pneumothorax (5%), heart failure (2%), pneumonia (1%), and gastrointestinal perforation (1%) were observed. These studies suggested that pazopanib can be considered as one of the treatment options showing improvement in progression-free survival rates in patients with most types of soft-tissue sarcoma. However, severe side-effects including pneumothorax, thrombocytopenia, and intestinal perforation must be taken into consideration.

Sorafenib is an oral multikinase inhibitor targeting mitogen-activated protein kinases (MAPK), PDGFRs, VEGFRs, and c-Kit [23]. Previous clinical studies have demonstrated the efficacy of sorafenib in several malignancies [24,25]. The Italian Sarcoma Group investigated the efficacy and safety of sorafenib in a phase II study with sorafenib for advanced osteosarcomas [26]. In this study, 35 patients who received 400 mg sorafenib twice daily, showed median progression-free survival and overall survival rates of 4 and 7 months, respectively. Three patients (8%) showed a partial response, two patients (6%) exhibited a minor response (tumor shrinkage < 30%), and 12 patients (34%) had stable disease. In another phase II study, the efficacy and safety of sorafenib was investigated in patients with soft-tissue sarcoma [27]. In this study, 101 patients with advanced soft-tissue sarcoma were given 400 mg sorafenib twice daily for 28 days. Eleven patients (15%) had a partial response, and 25 patients (33%) exhibited stable disease, with median progression-free survival and overall survival of 3 and 10 months, respectively.

In a phase II study, effects of sorafenib were evaluated in patients with recurrent or metastatic sarcoma [28]. In 37 patients with angiosarcoma, complete response was seen in one patient (3%), partial response was seen in four patients (11%), and stable disease was seen in 21 patients (57%). In 37 patients with leiomyosarcoma, one patient (3%) showed partial response and 18 (50%) showed stable disease. On the other hand, no objective response was seen in patients with malignant peripheral nerve sheath tumor (MPNST), synovial sarcoma, and undifferentiated pleomorphic sarcoma. In a retrospective study, efficacy of sorafenib in patients with desmoid tumors was evaluated [29]. In the study, 6/24 (25%) patients showed partial response and 17/24 (70%) showed stable disease. In a phase III study of sorafenib in desmoid tumors, significant improvement of progression-free survival has been observed in patients treated with sorafenib [30]. A phase II study with the combination of sorafenib and everolimus in patients with unresectable osteosarcoma was reported in 2015 by Italian Sarcoma Group [31]. In the study, 38 patients received 800 mg sorafenib plus 5 mg everolimus once a day until disease progression or unacceptable toxic effects. The study showed that 17 of 38 patients (45%) were progression free at 6 months. Further clinical studies with sorafenib are needed to obtain more information about the efficacy of sorafenib for advanced sarcomas.

*BCR-ABL* tyrosine kinase fusion gene arises due to an acquired (9;22) genetic translocation that causes chronic myelogenous leukemia and *BCR-ABL* is associated with proliferation of tumor cells and induces disease progression [32]. Imatinib mesylate, a tyrosine kinase inhibitor originally developed as an inhibitor of the BCR-ABL tyrosine kinase, was also found to inhibit c-Kit [33]. Expression of mutated form of c-Kit is usually observed in gastrointestinal stromal tumors (GISTs), and GISTs usually respond to imatinib mesylate [34]. Several studies investigating the effect of imatinib on soft-tissue tumors have been reported. In 2014, Ugurel et al. reported the safety and efficacy of imatinib assessed in a phase II study with imatinib in patients with dermatofibrosarcoma protuberans (DFSP) [35]. In 14 patients with DFSP, rates of complete response (CR), partial response (PR), stable disease (SD), and progressive disease (PD) were 7, 50, 36, and 7%, respectively. In 2017, the German Interdisciplinary Sarcoma Group reported the effects and safety of imatinib on patients with desmoid tumors evaluated in a phase II study [36]. In 38 patients who underwent imatinib treatment (800 mg, daily), progression-free survival rates at 6, 12, and 24 months were 65, 59, and 45%, respectively. A phase II study with imatinib (400 mg, daily) was performed in 13 patients with desmoplastic small round-cell tumors (DSRCTs) [37]. In the eight evaluated patients, SD in one patient and PD in seven patients were observed. The study indicated imatinib shows only a limited efficacy in patients with DSRCTs. Overall, these studies suggested that imatinib can be beneficial in some soft-tissue tumors, such as DFSP and desmoid tumors.

Olaratumab is a monoclonal antibody directed towards PDGFR. In randomized study, effect of olaratumab plus doxorubicin was compared with doxorubicin in patients with unresectable or metastatic soft tissue sarcoma [38]. The objective response rate was seen in 18% of patients with olaratumab plus doxorubicin and 12% with doxorubicin. Median overall survival was 26.5 months with olaratumab plus doxorubicin and 14.7 months with doxorubicin. Median progression-free survival was 6.6 months with olaratumab plus doxorubicin and 4.1 months with doxorubicin. Based on the results, olaratumab was conditionally approved by Food and Drug Administration (FDA), in combination with doxorubicin, as first-line treatment for soft tissue sarcoma (STS). Phase III study with olaratumab and doxorubicin in patients with advanced sarcomas is ongoing (NCT02451943). This study may determine the benefits of adding olaratumab to doxorubicin.

Rogorafenib is a multikinase inhibitor targeting tumor cells, vasculature, angiogenesis, and the tumor microenvironment by blocking the activity of several protein kinases, including those involved in the regulation of angiogenesis (VEGFR-1, VEGFR-2, and VEGFR-3, and TIE2), oncogenesis (KIT, RET, RAF-1, BRAF, and BRAFV600E), and tumor microenvironment (PDGFR and FGFR) [39]. In a phase II study in patients with metastatic osteosarcoma, 17 of 26 patients treated with rogorafenib were progression-free at 8 weeks compared with no patients in placebo group [40].

Cabozantinib is a small molecule inhibitor of multiple tyrosine kinases including MET, VEGF2, RET, and AXL. Cabozantinib was approved by FDA in 2012 for patients with metastatic thyroid cancer and advanced renal cell carcinoma [41]. A phase I study of cabozantinib in 41 patients with refractory or relapsed solid tumors, four cases including two medullary thyroid cancer, Wilms tumor, and clear cell sarcoma showed partial response, and seven cases including two medullary thyroid cancer, one Ewing’s sarcoma, one synovial sarcoma, one ASPS, one paraganlioma, and one ependymoma showed stable disease [41].

Exportin 1 (XPO1) is a critical mediator of nuclear export responsible for shuttling more than 200 known cargo proteins including tumor suppressor, anti-inflammatory, and growth-regulating proteins from the nucleus to the cytoplasm [42,43]. WPO1 is overexpressed in several malignancies and associated with poor outcomes [43,44,45,46,47,48]. Selinexor, orally bioavailable small molecule, inhibits XPO1 by binding cysteine-528 which is an essential residue for XPO1. XPO1 inhibition results in nuclear accumulation of p53, pRb, p21, p27, BRCA, FOXO, survivin, and other proteins [42,43]. Accumulation of tumor suppressor proteins in the nucleus induce growth arrest and apoptosis in tumor cells [49,50]. A phase I study of selinexor in patients with advanced bone and soft tissue sarcoma showed that none of the patients had objective response, 17 of 52 (33%) patients had stable disease at 4 months [51]. Especially, prolonged disease control was seen in patients with liposarcoma and leiomyosarcoma. A randomized study of selinexor versus placebo is currently ongoing in patients with dedifferentiated liposarcoma (ClinicalTrials.gov identifier NCT02606461).

TRC105 (carotuximab) is a monoclonal antibody to endoglin, an essential angiogenic target highly expressed on proliferating endothelium and tumor cells in angiosarcoma. TRC105 inhibits angiogenesis, tumor growth, and metastases in preclinical models and complements the activity of multi-kinase VEGFRI [52,53]. Currently, efficacy and safety of TRC105 combined with pazopanib have been evaluated in a phase III study in patients with advanced angiosarcoma [54].

Tenosynovial giant cell tumor, known as pigmented villonodular synovitis, is a locally-aggressive tumor. Although surgical resection is the standard treatment, complete resection is difficult in a part of patients with tenosynovial giant cell tumor. Tenosynovial giant cell tumor frequently have t(1;2) translocation that links the *CSF1* gene on chromosome 1p13 to the *COL6A3* gene on chromosome 2q35 [55], and often express colony-stimulating factor 1 (CSF1) [56]. In a phase I study of CSF1 receptor inhibitor PLX3397 in patients with tenosynovial giant cell tumor, 12 of 23 patients had a partial response and seven of 23 patients had stable disease [57]. This study suggests that signaling between CSF1 and CSF1 receptor can be therapeutic target for tenosynovial giant-cell tumors.

The neurotrophin tropomyosin receptor kinases TRKs are encoded by the neurotrophic receptor tyrosine kinase (*NTRK*) genes [58]. TRKs play a diverse role in neurobiology, but also are implicated in the pathogenesis of a subset of cancers with oncogenic fusions involving one of *NTRK* genes. NRK fusions have been reported in a diverse range of malignancies [59]. Larotrectinib, a selective inhibitor of TKR, has significant activity in TRK fusion malignancies [60]. In a phase I/II study of larotrectinib in patients with TRK fusion sarcomas, all of five patients had partial response [61]. Furthermore, three patients achieved complete (two patients) or near-complete (one patient) pathologic responses. These results indicate that larotrectinib can be a with current approaches, although further clinical study is required to assess the effect and safety in patients with TRK fusion sarcomas.

These studies suggest that targeting therapies may improve clinical outcomes in patients with bone and soft tissue tumors, although ongoing clinical trials of these targeting therapies should provide safeties and efficacies of those antitumor agents for bone and soft tissue sarcomas.

### 2.2. Eribulin

Eribulin is a synthetic analog of halichondrin B, a natural product isolated from marine sponge *Halichondria okadai* [62,63]. Eribulin inhibits microtubule polymerization through a specific binding site on β-tubulin and has tubulin-based antimitotic effect and disrupts mitotic spindle formation by arresting the cells in G2-phase and M-phase.

On the basis of preclinical activity observed in fibrosarcoma and leiomyosarcoma, EORTC/STBSG conducted a phase II study with eribulin in patients with recurrent or metastatic soft-tissue sarcoma [64]. In this study, progression-free survival rates at 12 weeks were 32% in patients with leiomyosarcoma, 47% in patients with adipocytic sarcoma, 21% in patients with synovial sarcoma, and 19% in patients with other sarcomas, respectively. In 2017, Kawai et al. reported a phase II study with eribulin in patients with advanced soft-tissue sarcoma [65]. In this study, 51 patients underwent treatment with eribulin mesylate (1.4 mg/m^2^) intravenously on days 1 and 8 over a 21-day cycle. Progression-free survival durations were 5.5 months in patients with liposarcoma/leiomyosarcoma, and 2.0 months in other sarcoma patients.

In 2016, EORTC reported a phase III study with eribulin in patients with non-operable or metastatic soft-tissue sarcoma subtypes of leiomyosarcoma and liposarcoma (NCT01327885) [66]. The study demonstrated a statistically significant improvement of two months in overall survival in patients treated with eribulin, compared with patients treated with dacarbazine, although no significant improvement in progression-free survival rates was observed.

Compared to pazopanib and trabectedin, only eribulin showed a significant improvement in overall survival in patients with advanced sarcomas. These previous studies suggest that eribulin is one of the treatment options in patients with advanced sarcomas, especially in patients with liposarcoma and leiomyosarcoma. However, further studies analyzing the effects of eribulin in each subtype of sarcoma are required.

### 2.3. Trabectedin

Trabectedin is a tetrahydroisoquinoline alkaloid that was initially isolated from the marine tunicate *Ecteinascidia turbinate* [67]. Major mechanism of trabectedin is related to binding of the minor groove of DNA and interference of late S-phase and G2-phase of cell cycle, and inhibition of the association of DNA-binding proteins [68].

EORTC/STBSG and the Sarcoma Alliance for Research through Collaboration group investigated the efficacy of trabectedin as first-line chemotherapy drug for advanced soft-tissue sarcoma. In the phase IIb study (TRUSTS trial), 133 patients were treated with doxorubicin (75 mg/m^2^ infusion on day 1, every three weeks) or trabectedin (1.5 mg/m^2^/24-h intravenous infusion on day 1 every three weeks) [69]. The median progression-free survival in patients treated with trabectedin and doxorubicin were 3 months and 6 months, respectively, and no significant improvement by trabectedin was seen in the study.

In a multicenter, randomized, open-label, phase II study, the safety and efficacy of trabectedin compared with the best supportive care in 76 patients with advanced translocation-related sarcomas were evaluated [70]. The study patients comprised of myxoid/round cell liposarcoma, synovial sarcoma, alveolar rhabdomyosarcoma, and Ewing sarcoma. In this study, trabectedin was intravenously administered in patients at the dose of 1.2 mg/m^2^ over 24 h, and significantly improved progression-free survival in patients with myxoid/round-cell liposarcoma and synovial sarcoma (5.6 months versus 0.9 months). However, no significant improvement in overall survival was observed in the trabectedin group.

In a multicenter, randomized, open-label, phase III study, effects of trabectedin were compared with dacarbazine in patients with advanced leiomyosarcoma and liposarcoma [71]. The study patients received intravenous trabectedin (1.5 mg/m^2^ over 24 h) or dacarbazine (1 g/m^2^, 20–120 min). In this study, significantly longer progression-free survival duration (4.2 months versus 1.5 months) was observed in patients who received trabectedin, although no significant difference in overall survival was observed between the two groups (12.4 months versus 12.9 months). In this study, elevated creatine kinase was observed in 5% of study patients, and 1.2% patients who received trabectedin developed rhabdomyolysis. Based on these results, trabectedin was approved by Food and Drug Administration for liposarcoma and leiomyosarcoma.

## 3. Immune Checkpoint Inhibitors

Recent basic and clinical research has demonstrated the association of immune checkpoints with progression of malignancies, and efficacy of immune checkpoint inhibitors on various malignancies [72,73,74,75,76,77]. Basic research and clinical studies on immune checkpoint inhibitors have been demanded for their application in the treatment of sarcomas. However, immune-related adverse events due to the activation of immune system by immune checkpoint blockade are very likely to occur [78]. Immune-related adverse events which include inflammatory side-effects of immune checkpoint inhibitors, can affect gastrointestinal tract, endocrine glands, skin, liver, and central nervous, cardiovascular, pulmonary, musculoskeletal, and hematologic systems. Although most of the immune-related adverse events are reversible, a part of the events might be permanent and could lead to deaths.

Engagement of programmed cell death 1 (PD-1) receptor on T-cells and programmed death-ligand 1 (PD-L1) is a key mechanism of immune system-mediated escape of malignancies. In a meta-analysis study, PD-L1 expression significantly associated with overall survival in patients with bone sarcomas (osteosarcoma and chondrosarcoma) and event-free survival in patients with bone and soft-tissue sarcoma [79]. In another meta-analysis study, overexpression of PD-L1/PD-1 significantly correlated with metastasis in patients with osteosarcoma, although no significant association of PD-1/PD-L1 expression was observed with overall survival [80]. Similarly, a study for Ewing’s sarcoma showed that metastatic tumors showed higher expression of PD-L1, although no significant association was observed between PD-L1 expression and progression-free survival or overall survival [81]. These studies suggested that immune checkpoints can be therapeutic target for the treatment of sarcomas.

Recently, immune checkpoint inhibitors for sarcomas were investigated in several clinical studies (Table 2) [82,83,84,85,86]. In 2017, Tawbi et al. reported a single-arm phase II study of pembrolizumab, an anti-PD-1 antibody [85]. In this study, activity and safety of pembrolizumab were assessed in 80 patients with advanced bone and soft-tissue sarcoma. In 40 patients with soft-tissue sarcoma, seven patients (18%) showed objective response, including four of 10 patients (40%) with undifferentiated pleomorphic sarcoma, two of 10 patients (20%) with liposarcoma, and one of 10 patients (10%) with synovial sarcoma. In 40 patients with bone sarcoma, 1 of 22 patients (5%) with osteosarcoma and one of five patients (20%) with chondrosarcoma showed an objective response. No patients with Ewing’s sarcoma exhibited an objective response. Treatment-emergent serious adverse events (SAEs) were observed in nine of 80 patients (11%), and immune-related SAEs, including adrenal insufficiency (2%), pneumonitis (2%), and nephritis (1%), were also observed. In 2018, Toulmonde et al. reported a phase II study with pembrolizumab in patients with advanced soft-tissue sarcoma [86]. In this study, 6-month progression-free rates were 0% in patients with leiomyosarcoma, 0% in patients with undifferentiated pleomorphic sarcoma, and 14.3% in patients with other sarcomas, respectively.

Although monotherapy with immune checkpoints inhibitor has shown a limited effect in the previous reports, a study with combination therapy, such as CTLA-4 inhibitor and PD-1 inhibitor suggested that combination therapy has a synergistic effect in the mouse model of melanoma [87]. Furthermore, the efficacy of combination therapy has been investigated in clinical trials for various types of malignancies [88,89,90]. A phase II study with nivolumab, with or without ipilimumab in patients with metastatic sarcoma was reported by D’Angelo et al. [91]. In this study, 85 patients with metastatic sarcoma underwent nivolumab monotherapy (3 mg/kg) or combined therapy with nivolumab and ipilimumab (1 mg/kg). Response rate for patients treated with nivolumab was 5% (2 of 38 patients), while response rate for patients treated with a combination of nivolumab and ipilimumab was 16% (6 of 38 patients). This study suggests that combined immune checkpoint inhibition is a promising method for the treatment of sarcomas. Further clinical studies are needed to assess the efficacy and the adverse events related to the combination therapy. Furthermore, biomarkers that distinguish between responders and non-responders on immunotherapy are urgently needed.

## 4. Cellular Immunotherapy

In 1891, the first study concerning immunotherapy was reported by Coley [92]. Coley’s study involving injection of live or inactivated *Streptococcus pyogenes* and *Serratia marcescens* (Coley’s toxin) showed complete response in 10% of patients with unresectable sarcoma. Although efficacies of various vaccines including *Bacillus Calmette-Guerin* (BCG), allogenic tumor cells, muramyl tripeptide (MTP) have been investigated, no significant improvement was observed than in the previous studies [93,94,95,96]. There are reports investigating the efficacy of application of cellular immunotherapy for the treatment of sarcomas (Table 3) [97,98,99,100,101].

Dendritic cells, the representative antigen-presenting cells, identify tumor-specific antigens via major histocompatibility complex (MHC) class I, and have the ability to activate tumor immune response by presenting the antigens to the other immune cells. Since dendritic cells play a pivotal role in tumor immune system, dendritic cells have been considered as a promising adjuvant for inducing an antitumor immune response [102]. Although dendritic cells-based immunotherapy has shown low incidence of adverse events in patients with sarcomas [97,103,104], previous studies have shown benefits of dendritic cell-based immunotherapy in only a small percentage of patients [102].

On the other hand, recent studies suggest that immunotherapy utilizing tumor antigen-specific T-cells is a promising treatment option in patients with sarcoma. Adoptive transfer of T-cells, genetically modified to express chimeric antigen receptors (CARs), showed a high rate of tumor regression in patients with leukemia [105,106]. CARs are transmembrane molecules composed of several functional parts. An extracellular single-chain variable fragment is fused to a hinge/spacer module and a transmembrane domain. The transmembrane domain is linked to the intracellular domain, which is critical for transmission of the activation signal. T-cells expressing CARs can specifically recognize the target antigen and eliminate the target tumor cells. In a phase I/II clinical study using T-cells expressing human epidermal growth factor 2 (HER2)-specific chimeric antigen receptor (HER2-CAR T-cells), 19 patients with HER2-positive sarcomas underwent HER2-CAR T-cell therapy [100]. In this study, four of 17 evaluated patients had stable disease for 12 weeks to 14 months, and ≥90% tumor necrosis was observed in three patients who showed stable disease. No significant adverse event was observed in the study.

Cancer testis antigens have emerged as both a diagnostic marker and a therapeutic target in malignant tumors. Cancer testis antigen NY-ESO-1 (New York esophageal squamous cell carcinoma 1) is encoded by the *CTAG1B* gene, and expressed in 10–50% of metastatic melanomas, breast, prostate, thyroid, and ovarian cancers [107,108,109,110]. Since it is reported that expression of NY-ESO-1 was seen in 70–80% of synovial sarcomas, but not in any normal adult tissues except the testis [111,112,113], NY-ESO-1 has been considered as an attractive target of immune-based therapy. In the study using transferred autologous T-cells transduced with a T-cell receptor (TCR) directed against NY-ESO-1, clinical response was observed in four of six patients (67%) with synovial sarcoma and five of 11 patients (45%) with melanoma bearing tumors-expressing NY-ESO-1 [114]. No severe toxicity was attributed to the transferred cells in the study. In 2018, D’Angelo et al. reported a study with genetically engineered-lymphocytes reactive with NY-ESO-1 [115]. In this study, patients with NY-ESO-1-positive synovial sarcoma and melanoma underwent treatment with autologous TCR-transduced T-cells and interleukin-2. TCR recognizing NY-ESO-1 was transduced by a retroviral vector. Objective responses were seen in 67% (four of six patients) of synovial sarcoma, and 45% (five of 11 patients) of melanoma. No toxicities were observed due to transferred cells, although all the patients had transient neutropenia and thrombocytopenia, induced by IL-2 treatment.

These studies suggest that adoptive T-cell immunotherapies are a promising treatment option with high response rate. However, significant toxicity related to adoptive T-cell immunotherapy was reported in various malignancies [116,117,118,119]. Cytokine release syndrome (CRS), and adverse events related to adoptive T-cell therapy are thought to be an elevated systemic inflammatory response due to the activation of CAR-T-cells. CRS ranges from mild (fever, fatigue, and mild hypotension) to severe symptoms, including severe hypotension, respiratory failure, and multi-organ failure. Although severe adverse events have been reported in studies with adoptive T-cell therapy, previous studies have shown significant benefits in patients with advanced sarcomas. Further investigation is required to test the efficacy and safety of TCR therapy and CAR-T-cell therapy.

## 5. Future Directions

Due to the rarity and varieties of sarcomas, information about antitumor agents for each type of tumor is not satisfactorily available. To select the appropriate treatment options for each tumor, further basic and clinical research for each tumor is required. On the other hand, malignant tumors have diversified gene mutations, and the gene mutations and expression of various proteins, such as PD-L1 are different between primary and metastatic tumors. Therefore, novel customized systemic treatments, such as immunotherapy using patient-derived tumor lysate for each patient [102], and patient-derived orthotopic xenograft nude mouse model [120,121] are anticipated for the improvement of advanced sarcomas.

## 6. Conclusions

Although systemic treatments using anticancer agents have improved outcomes in bone and soft-tissue sarcomas, doxorubicin-based regimens remain the standard treatment for sarcomas. Numerous clinical studies have not been able to demonstrate the advantage of new anticancer agents over doxorubicin. Further investigation into systemic management of sarcomas is needed to improve the outcomes in patients with sarcomas. Since immune checkpoints inhibitors, such as nivolumab and ipilimumab demonstrated significant benefits in several malignancies, immune system, and tumor microenvironment have become promising candidates for therapeutic targeting of bone and soft-tissue sarcomas. Further basic and clinical studies for mechanisms of regulation of the tumor microenvironment, new treatment agents, and combination of anticancer agents are needed which may contribute to the development of sarcoma treatments.

## Figures and Tables

**Table 1 ijms-20-00170-t001:** Clinical studies and target molecules for bone and soft tissue tumors.

Treatment	Target Molecule	N	Phase	Tumor Type	Clinical Significance	Grade 3-4 Toxicities	References
Pazopanib (daily, 800 mg)	VEGF-1, 2, 3 PDGF-α, c-Kit	142	Phase II	STS	PFS at 12 weeks: 44% in LMS, 49% in SS, and 39% in other sarcomas	hyperbilirubinemia (6%), hypertension (8%), and fatigue (8%)	[19]
Pazopanib (daily, 800 mg) or placebo	VEGF-1, 2, 3 PDGF-α, c-Kit	372	Phase III	metastatic sarcoma	OS: 12.5 (pazopanib) and 10.7 (placebo) months PFS: 4.6 (pazopanib) and 1.6 (placebo) months)	fatigue (13%), diarrhea (5%), nausea (3%), hypertension (7%), anorexia (6%), vomiting (3%), rash (< 1%), and mucositis (1%)	[20]
Pazopanib (daily, 800 mg)	VEGF-1, 2, 3 PDGFR, c-Kit	41	Phase II	liposarcoma	PFS: 4.4 months OS: 12.6 months	3 deaths: possible complication of the treatment	[21]
Sorafenib (400 mg, (twice, daily)	MAPK, PDGFRs, VEGFRs, and c-Kit	35	Phase II	advanced osteosarcoma	median PFS and OS were 4 and 7 months	anemia (6%), thrombocytopenia (6%), nausea (3%), lipase elevation (3%), abdominal cramps (3%), oral mucositis (3%), skin reaction (3%), and skin bleeding after trauma (3%)	[26]
Sorafenib (400 mg, twice, daily for 28 days)	MAPK, PDGFRs, VEGFRs, and c-Kit	101	Phase II	STS	PR 15%, SD 33% Median PFS: three months Median OS: 10 months	diarrhea (7%), fatigue (5%), hand-foot syndrome (4%), rash (4%), anorexia (2%), emesis (2%), hypophosphataemia (1%), and myalgia (1%).	[27]
Imatinib (800 mg, daily)	PDGFR and c-Kit	38	Phase II	Desmoid tumor	Response rate 19%	Grade 4 (neutropenia): 3% Grade 3 (neutropenia, leucopenia, nausea, vomiting, gastritis, rash, and contracture): 11%	[36]
Olaratumab (15 mg/kg, day 1 and day 8 plus doxorubicin 75 mg/m^2^ or doxorubicin alone)	PDGFR	133	Phase II	STS	Median OS: 27 months with olaratumab plus doxorubicin and 15 months with doxorubicin Objective response: 18% with olaratumab plus doxorubicin and 12% with doxorubicin	Grade 3: 13% with olaratumab plus doxorubicin and 17% with doxorubicin Grade ≥ 4: 9% with olaratumab plus doxorubicin and 8% with doxorubicin	[38]

VEGFR, vascular endothelial growth factor receptor; PDGFR, platelet-derived growth factor receptor; MAPK, mitogen-activated protein kinases; STS, soft tissue sarcoma; PFS, progression-free survival; OS, overall survival, LMS, leiomyosarcoma; SS, synovial sarcoma.

**Table 2 ijms-20-00170-t002:** Clinical studies of immune checkpoint inhibitors.

Treatment	N	Design	Tumor Type	Clinical Significance	Grade 3-4 Toxicities	References
Pembrolizumab (200 mg, iv, every 3 weeks)	86	Phase II	Bone and soft tissue sarcoma	Response rates: UPS (40%), liposarcoma (20%), SS (10%), LMS (0%), osteosarcoma (5%), chondrosarcoma (20%), and ES (0%),	pulmonary embolism (1%), Adrenal insufficiency (1%), interstitial nephritis (1%), Infectious pneumonia (1%), bone pain (1%), pleural effusion (1%), and hypoxia (1%),	[85]
Cyclophosphamide (50 mg, twice, daily, 1 week on and 1 week off) and Pembrolizumab (200 mg, iv, every 3 weeks)	57	Phase II	STS and GIST	Objective response: 2% six-month PFS rate: LMS (0%), UPS (0%), GIST (11%), others (14%) Median PFS: 1.4 months	Fatigue (3.5%), oral mucositis (3.5%), anemia (7%), and lymphocytopenia (35%)	[86]
Ipilimumab (1–10 g/m^2^, iv, every 3 weeks)	33	Phase I	Melanoma, sarcoma, and refractory solid tumors	CR (0%), PR (0%), SD (18%), and PD (82%)	Colitis/diarrhea (9%), transaminitis (6%), endocrinopathies (3%), and others (9%)	[82]
Ipilimumab 10 or 3 mg/kg, every 3 weeks) and dasatinib (70 or 100 mg daily, or 70 mg twice daily)	28	Phase I	GIST and sarcomas	Median PFS: 2.8 months, median OS: 13.5 months CR (0%), PR (0%), SD (11%), and PD (89%)	Anemia (21%), lymphopenia (13%), diarrhea (4%), edema (4%), infection (4%), nausea (4%), pericardial effusion (4%), and vomiting (4%)	[83]
Nivolumab (3 mg/kg, iv, every 2 weeks)	12	Phase II	LMS of uterus	Objective response: 0% Median PFS: 1.8 months	Abdominal pain (8%), increased amylase (8%), increased lipase (8%), and fatigue (8%)	[84]

STS, soft tissue sarcoma; GIST, gastrointestinal stromal tumors; LMS, leiomyosarcoma; UPS, undifferentiated pleomorphic sarcoma; SS, synovial sarcoma; ES, Ewing’s sarcoma; PFS, progression-free survival; CR, complete response; PR, partial response; SD, stable disease; PD, progression disease.

**Table 3 ijms-20-00170-t003:** Recent clinical studies of cellular immunotherapy.

Immunotherapy	N	Design	Tumor Type	Treatment	Clinical Significance	Grade 3-4 Toxicities	References
DCs	37	Phase I/2	Bone and soft tissue sarcoma	DCs pulsed with TL/TNFα/OK-432	CR 0%, PR 3%, SD 17%, and PD 80%	None	[97]
DCs	43	Phase I/2	EWS, RMS, and NB	29 patients: immunotherapy using autologous lymphocytes, TL-pulsed DCs, with or without IL7	5-year OS in patients with or without response to immunotherapy: 73% and 37%	Transaminitis (7%), fever (4%), and anaphylaxis (4%), attributed to IL7	[98]
DCs	15	Phase I/2	NB, EWS, osteosarcoma, and RMS	Decitabine followed by DC pulsed with MAGE-A1, MAGE-A3 and NY-ESO-1 peptides	CR: 10%, SD 20%, PD 70%	Neutropenia 40%, myelotoxity 10%, elevated ALP 10%, increased ALT 10%	[99]
HER2-CAR T cells	19	Phase I/2	Osteosarcoma, EWS, RMS, PNET, DSRCT	HER2-CAR T cells	SD 24%	None	[100]
NY-ESO-1 TCR T cells	38	Phase II	Melanoma, SS	NY-ESO-1 TCR T cells and IL-2	CR 14% PR 46%	Neutropenia and thrombocytopenia associated with IL2 (100%)	[101]

DC, dendritic cell, HER2, human epidermal growth factor receptor 2; CAR, chimeric antigen receptor; TCR, T cell receptor; EWS, Ewing’s sarcoma; RMS, rhabdomyosarcoma; NB, neuroblastoma; PNET, primitive neuroectodermal tumor; DSRCT, desmoplastic small round cell tumor; SS, synovial sarcoma; TL, tumor lysate; TNFα, tumor necrosis factor α; IL, interleukin; MAGE, CR, complete response; PR, partial response; SD, stable disease; PD, progression disease; OS, overall survival.

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
