# Peer review of "Therapeutic Targets for Bone and Soft-Tissue Sarcomas"

_ijms, 2019, doi:10.3390/ijms20010170_

Reviewer 1 Report

The authors provide a concise and readable review on the current and future treatment options for a group of rare and heterogeneous tumors; the bone and soft-tissue sarcomas. Particularly the discussion of immunotherapeutic options for sarcomas is timely and of interest. The authors should address the following comments to improve their manuscript Major comments: 1. A important advancement in the treatment of soft-tissue sarcomas is missing from the current manuscript and should be included.  The doxorubicin-based combination with olaratumab may have a huge impact on the outcome of advanced/metastatic STS patients (Tap W.D. et al. (2016) Lancet 388:488-497). Olaratumab, a monoclonal antibody directed towards PDGFR-a, was conditionally approved, in combination with doxorubicin, as first-line treatment for STS. 2. In section 2.2 which deals with sorafenib a few important references are missing: Maki, R.G. et al. (2009) J Clin Oncol. 27(19):3133-3140 and Gounder M.M. et al. (2011) Clin Cancer Res 17(12):4082-). In particular the latter reference is of interest as the preliminary results of an ongoing phase III trial investigating sorafenib in desmoid tumors appear to show  efficacy  as was presented at the last ASCO meeting.  Please incorporate and discuss these references. 3. In section 3 it may be informative to include a remark that biomarkers that distinguish between responders and non-responders on immunotherapy are urgently needed. Minor comments: 1. Table 1 – In the column listing the target molecules c-Kit is referred to in various ways (c-kit, KIT, c-Kit tyrosine kinase). Please be consistent. Note that pazopanib targets VEGFR-1, VEGFR-2 and VEGFR-3 (not VEGF-1,2,3) and PDGFRA (not PDGF-α). Please check and correct. 2. Page 4, line 107 – Is a tumor shrinkage of

Author Response

Major comments:

1. A important advancement in the treatment of soft-tissue sarcomas is missing from the current manuscript and should be included. The doxorubicin-based combination with olaratumab may have a huge impact on the outcome of advanced/metastatic STS patients (Tap W.D. et al. (2016) Lancet 388:488-497). Olaratumab, a monoclonal antibody directed towards PDGFR-a, was conditionally approved, in combination with doxorubicin, as first-line treatment for STS.

Response:Thank you for your suggestion. The following sentences describing olaratumab were added as section 2.6 Olaratumab. Olaratumab is a monoclonal antibody directed towards PDGFR-a. In randomized study, effect of olaratumab plus doxorubicin was compared with doxorubicin in patients with unresectable or metastatic soft tissue sarcoma [Tap WD Lancet, 2016]. The objective response rate was seen in 18% of patients with olaratumab plus doxorubicin and 12% with doxorubicin. Median overall survival was 26.5 months with olaratumab plus doxorubicin and 14.7 months with doxorubicin. Median progression-free survival was 6.6 months and 4.1 months with doxorubicin. Based on the results, olaratumab was conditionally approved by FDA, in combination with doxorubicin, as first-line treatment for STS.

2. In section 2.2 which deals with sorafenib a few important references are missing: Maki, R.G. et al. (2009) J Clin Oncol. 27(19):3133-3140 and Gounder M.M. et al. (2011) Clin Cancer Res 17(12):4082-). In particular the latter reference is of interest as the preliminary results of an ongoing phase III trial investigating sorafenib in desmoid tumors appear to show efficacy as was presented at the last ASCO meeting. Please incorporate and discuss these references.

Response: Thank you for your suggestion. The following sentences were added to section 2.2. In a phase 2 study, effects of sorafenib were evaluated in patients with recurrent or metastatic sarcoma [Maki RG, J Clin Oncol, 2009]. In 37 patients with angiosarcoma, complete response was seen in 1 patient (3%), partial response was seen in 4 patients (11%), and stable disesase was seen in 21 patients (57%). In 37 patients with leiomyosarcoma, 1 patient (3%) showed partial response and 18 (50%) showed stable disease. On the other hand, no objective response was seen in patients with MPNST, synovial sarcoma, and undifferentiated pleomorphic sarcoma.  

In a retrospective study, efficacy of sorafenib in patients with desmoid tumors was evaluated [Gounder MM, Clin Cancer Res, 2011]. In the study, 6/24 (25%) patients showed partial response and 17/24 (70%) showed stable disease.

In a phase 3 study of sorafenib in desmoid tumors, significant improvement of progression free survival has been observed in patients treated with sorafenib [Gounder MM. J Clin Oncol, 2018].

References:

1. Maki RG, J Clin Oncol, 2009

2.  Gounder MM, Clin Cancer Res, 2011

3.  Gounder MM, Mahoney MR, Van Tine BA, et al. Phase III, randomized, double blind, placebo-controlled trial of sorafenib in desmoid tumors (Alliance A091105). J Clin Oncol 36: Supp.11500, 2018

3. In section 3 it may be informative to include a remark that biomarkers that distinguish between responders and non-responders on immunotherapy are urgently needed.

Response: Thank you for your suggestion. The following sentence was added to the section 3. Biomarkers that distinguish between responders and non-responders on immunotherapy are urgently needed.

Minor comments: 1. Table 1 – In the column listing the target molecules c-Kit is referred to in various ways (c-kit, KIT, c-Kit tyrosine kinase). Please be consistent. Note that pazopanib targets VEGFR-1, VEGFR-2 and VEGFR-3 (not VEGF-1,2,3) and PDGFRA (not PDGF-α). Please check and correct.

Response: Thank you for your pointing these out. The table 1 was corrected according to the reviewer’s comments.

Reviewer 2 Report

In the abstract the authors state that the goal of their review is to summarize therapies and treatments for Bone and soft tissue sarcomas. The authors have broken up the review into six sections. In the introduction the authors explain why new therapies are needed to treat patients with sarcomas. The next three sections cover many clinical trials of small molecules and immunotherapies. The paper concludes that none of the clinical trials covered in the paper offer a better outcome than doxorubicin. In some of the sections there isn’t much background into the therapies, why they are used, how they relate to important pathways in sarcoma progression and resistance, or why they may have failed. Why do the authors include all of the toxicity results from each of the trials in their review? Is specific toxicity data important when considering therapies for patients with sarcomas?

Specific Comments: 

Section 2 describes a therapy and then lists clinical trials, responses and all of the side effects. It would be more useful to focus on the important results from these clinical trials and compare which drugs work best for each type of sarcoma. The Trabectedin section describes two clinical trials in detail in which no significant improvement was observed. In the third clinical trial significantly longer progression free survival was observed but not overall survival. From the description, it seems that the mechanism for Trabectedin is not well known? Why did the authors include all this information in the review article?

How is NY-ESO connected to synovial sarcoma? Is synovial sarcoma different than other sarcomas? Does NY-ESO only play a role in synovial sarcoma?

Author Response

Dear reviewer, we sincerely thank you for your considering our manuscript entitled “Current therapeutic targets for bone and soft-tissue sarcomas” for publication in International Journal of Molecular Sciences. We are very grateful for your prompt attention and thorough review. Based on the reviewers’ comments, we have revised the manuscript and have addressed all of the concerns brought up. Please see below for our point-by point responses to each of the reviewers’ comments. If there are any further issues that need to be resolved to improve our manuscript, please let us know.

Newly added or changed parts are indicated in red in the revised manuscript.

In the abstract the authors state that the goal of their review is to summarize therapies and treatments for Bone and soft tissue sarcomas. The authors have broken up the review into six sections. In the introduction the authors explain why new therapies are needed to treat patients with sarcomas. The next three sections cover many clinical trials of small molecules and immunotherapies. The paper concludes that none of the clinical trials covered in the paper offer a better outcome than doxorubicin. In some of the sections there isn’t much background into the therapies, why they are used, how they relate to important pathways in sarcoma progression and resistance, or why they may have failed. Why do the authors include all of the toxicity results from each of the trials in their review? Is specific toxicity data important when considering therapies for patients with sarcomas?

Response: Thank you for your comments. The background was added to each section. The toxicity results except for specific toxicity data were deleted from the manuscript.

Specific Comments:

Section 2 describes a therapy and then lists clinical trials, responses and all of the side effects. It would be more useful to focus on the important results from these clinical trials and compare which drugs work best for each type of sarcoma. The Trabectedin section describes two clinical trials in detail in which no significant improvement was observed. In the third clinical trial significantly longer progression free survival was observed but not overall survival. From the description, it seems that the mechanism for Trabectedin is not well known? Why did the authors include all this information in the review article?

Response: Thank you for your suggestion.

As described in section 5, information of information about antitumor agents for each type of tumor is not satisfactorily available. Therefore, further basic and clinical research for each tumor is required to select the appropriate treatment options for each tumor type. Trabectedin significantly improved progression free survival but could not improve overall survival. The reason of the discrepancy remains unclear.

How is NY-ESO connected to synovial sarcoma? Is synovial sarcoma different than other sarcomas? Does NY-ESO only play a role in synovial sarcoma?

Response: Thank you for pointing this out. Endo M et al. reported expression of NY-ESO-1 in each type of mesenchymal tumors. Among mesenchymal tumors, myxoid liopsarcomas showed the highest positivity for NY-ESO-1 (88%), followed by synovial sarcomas (49%), myxofibrosarcomas (35%), and conventional chondrosarccomas (28%). Lai et al. reported that synovial showed the highest positivity for NY-ESO-1 (82%), followed by dermatofibrosarcoma protuberans (10%), and angiosarcoma (10%). Since expression of NY-ESO-1 is not specific for synovial sarcoma, NY-ESO-1 can be promising therapeutic target against a part of sarcomas including synovial sarcoma and myxoid liposarcoma.

References:

1.   Endo M, de Graaff MA, Ingram DR, et al. NY-ESO-1 (CTAG1B) expression in mesenchymal tumors. Modern Pathol 2015

2.   Lai JP, Robbins PF, Raffeld M, Aung PP, Tsokos M, Rosenberg SA, et al. NY-ESO-1 expression in synovial sarcoma and other mesenchymal tumors: significance for NY-ESO-1-based targeted therapy and differential diagnosis. Mod Pathol 2012;25:854–8.

Reviewer 3 Report

This manuscript by Miwa et al provides a review on current therapies for bone and soft tissue sarcomas.

The main issue with this manuscript is the limited number of targets and drugs discussed. Far too much detail is provided on therapies that are already established, without enough discussion of new targets for treatment. For example, several promising drugs in phase I and II trials were not mentioned at all, including: regorafenib and cabozantinib (bone sarcomas), olaratumab (soft tissue sarcoma), selinexor (liposarcoma), anlotinib (alveolar soft parts sarcoma), TRC105 (angiosarcoma), pexdartinib (PVNS), pembrolizumab + axitinib (ASPS), and larotrectinib (NTRK-positive sarcomas). A discussion of these exciting new agents, and their respective molecular targets, must be included for the review to be relevant and contemporary.

Author Response

Dear reviewer, we sincerely thank you for your considering our manuscript entitled “Current therapeutic targets for bone and soft-tissue sarcomas” for publication in International Journal of Molecular Sciences. We are very grateful for your prompt attention and thorough review. Based on the reviewers’ comments, we have revised the manuscript and have addressed all of the concerns brought up. Please see below for our point-by point responses to each of the reviewers’ comments. If there are any further issues that need to be resolved to improve our manuscript, please let us know.

Newly added or changed parts are indicated in red in the revised manuscript.

This manuscript by Miwa et al provides a review on current therapies for bone and soft tissue sarcomas.

The main issue with this manuscript is the limited number of targets and drugs discussed. Far too much detail is provided on therapies that are already established, without enough discussion of new targets for treatment. For example, several promising drugs in phase I and II trials were not mentioned at all, including: regorafenib and cabozantinib (bone sarcomas), olaratumab (soft tissue sarcoma), selinexor (liposarcoma), anlotinib (alveolar soft parts sarcoma), TRC105 (angiosarcoma), pexdartinib (PVNS), pembrolizumab + axitinib (ASPS), and larotrectinib (NTRK-positive sarcomas). A discussion of these exciting new agents, and their respective molecular targets, must be included for the review to be relevant and contemporary.

Response: Thank you for your suggestion. The following sentences and references were added to the manuscript.

Rogorafenib is a multikinase inhibitor targeting tumor cells, vasculature, angiogenesis, tumor microenvironment by blocking the activity of several protein kinases, including those involved in the regulation of angiogenesis (VEGFR-1, VEGFR-2, and VEGFR-3, and TIE2), oncogenesis (KIT, RET, RAF-1, BRAF, and BRAFV600E), and tumor microenvironment (PDGFR and FGFR) [Wilhelm SM, Dumas J, Adnane L, Int J Cancer, 2011]. In a phase 2 study in patients with metastatic osteosarcoma, 17 of 26 patients treated with rogorafenib were progression-free at 8 weeks compared with no patients in placebo group [Duffaud, Lancet Oncol, 2018].

Cabozantinib is a small molecule inhibitor of multiple tyrosine kinases including MET, VEGF2, RET, and AXL. Cabozantinib was approved by FDA in 2012 for patients with metastatic thyroid cancer and advanced renal cell carcinoma [Chuk MK. Pediatr Blood Cancer. 2018]. A phase 1 study of cabozantinib in 41 patients with refractory or relapsed solid tumors, 4 cases including 2 medullary thyroid cancer, Wilms tumor, and clear cell sarcoma showed partial response, and 7 cases including 2 medullary thyroid cancer, 1 Ewing sarcoma, 1 synovial sarcoma, 1 ASPS, 1 paraganlioma, and 1 ependymoma showed stable disease [Chuk MK, Pediatr Blood Cancer, 2018].

Exportin 1 (XPO1) is a critical mediator of nuclear export responsible for shuttling more than 200 known cargo proteins including tumor suppressor, anti-inflammatory, and growth-regulating proteins from the nucleus to the cytoplasm [Fung HY, Semn Cancer Biol, 2014; Tan DS, Cancer Discov, 2014]. WPO1 is overexpression in several malignancies and associated with poor outcomes [Tan DS, Cancer Discov, 2014; Huang WY, Clin Invest Med, 2009; Noske A, Cancer 2008; Shen A Neurosurg, 2009; van der Watt PJ, Int J Cancer, 2019; Yao Y. Oncol Rep, 2009]. Selinexor, orally bioavailable small molecule, inhibits XPO1 by binding cysteine-528 which is an essential residue for XPO1. XPO1 inhibition results in nuclear accumulation of p53, pRb, p21, p27, BRCA, FOXO, surviving, and other proteins [Fung HY, Semn Cancer Biol, 2014; Tan DS, Cancer Discov, 2014]. Accumulation of tumor suppressor proteins in the nucleus induce growth arrest and apoptosis in tumor cells [Lapalombella R. Blood, 2012; Yoshimura M. Cancer Sci, 2014]. A phase 1 study of selinexor in patients with advanced bone and soft tissue sarcoma showed that none of the patients had objective response, 17 of 52 (33%) patients had stable disease at 4 months [Gounder MM, 2016, J Clin Oncol]. Especially, prolonged disease control was seen in patients with liposarcoma and leiomyosarcoma. A randomized study of selinexor versus placebo is currently ongoing in patients with dedifferentiated liposarcoma (ClinicalTrials.gov identifier NCT02606461).

TRC105 (carotuximab) is a monoclonal antibody to endoglin, an essential angiogenic target highly expressed on proliferating endothelium and tumor cells in angiosarcoma. TRC105 inhibits angiogenesis, tumor growth, and metastases in preclinical models and complements the activity of multi-kinase VEGFRI [Duffy AG, Clin Cancer Res 2017; Tian H, Faseb J 2018].

Currently, efficacy and safety of TRC105 combined with pazopanib have been evaluated in a phase 3 study in patients with advanced angiosarcoma [Mehta CR, 2018].  

Tenosynovial giant cell tumor, known as pigmented villonodular synovitis, is locally aggressive tumor. Although surgical resection is the standard treatment, complete resection is difficult in patients with tenosynovial giant cell tumor. Tenosynovial giant cell tumor frequently have t(1;2) translocation that links the CSF1 gene on chromosome 1p13 to the COL6A3 gene on chromosome 2q35 [Cupp JS. Am J Surg Pathol, 2007], and often express colony-stimulating factor 1 (CSF1) [Molena B. Clin Exp Rheumatol, 2011]. In a phase 1 study of CSF1 receptor inhibitor PLX3397 in patients with tenosynovial giant cell tumor, 12 of 23 patients had a partial response and 7 of 23 patients had stable disease [Tap WD N Engl J Med, 2015]. This study suggests that signaling between CSF1 and CSF1 receptor can be therapeutic target for tenosynovial giant-cell tumors.

The neurotrophin tropomyosin receptor kinases TRKs are encoded by the neurotrophic receptor tyrosine kinase (NTRK) genes [Skaper SD. Methods Mol Biol, 2012]. TRKs play a diverse role in neurobiology, but also are implicated in the pathogenesis of a subset of cancers with oncogenic fusions involving one of NTRK genes. NRK fusions have been reported in a diverse range of malignancies [Davis JL. Pediatr Dev Pathol, 2018]. Larotrectinib, an selective inhibitor of TKR, has significant activity in TRK fusion malignancies [Drilon A. N Engl J Med, 2018]. In a phase 1/2 study of larotrectinib in patients with TRK fusion sarcomas, all of 5 patients had partial response [DuBois SG. Cancer, 2018]. Furthermore, three patients achieved complete (2 patients) or near-complete (1 patient) pathologic responses. These results indicate that larotrectinib can be a with current approaches, although further clinical study is required to assess the effect and safety in patients with TRK fusion sarcomas.

References:

1.   Chuk MK, Widemann BC, Minard CG, Liu X, Kim A, Bernhardt MB, Kudgus RA, Reid JM, Voss SD, Blaney S, Fox E, Weigel BJ. A phase 1 study of cabozantinib in children and adolescents with recurrent or refractory solid tumors, including CNS tumors: Trial ADVL1211, a report from the Children's Oncology Group. Pediatr Blood Cancer. 2018 Aug;65(8):e27077.

2.   Duffaud F, Mir O, Boudou-Rouquette P, Piperno-Neumann S, Penel N, Bompas E, Delcambre C, Kalbacher E, Italiano A, Collard O, Chevreau C, Saada E, Isambert N, Delaye J, Schiffler C, Bouvier C, Vidal V, Chabaud S, Blay JY; French Sarcoma Group. Efficacy and safety of regorafenib in adult patients with metastatic osteosarcoma: a non-comparative, randomised, double-blind, placebo-controlled, phase 2 study. Lancet Oncol. 2018 Nov 23. pii: S1470-2045(18)30742-3. doi: 10.1016/S1470-2045(18)30742-3. [Epub ahead of print]

3.    Tap WD, Jones RL, Van Tine BA, Chmielowski B, Elias AD, Adkins D, Agulnik M, Cooney MM, Livingston MB, Pennock G, Hameed MR, Shah GD, Qin A, Shahir A, Cronier DM, Ilaria R Jr, Conti I, Cosaert J, Schwartz GK. Olaratumab and doxorubicin versus doxorubicin alone for treatment of soft-tissue sarcoma: an open-label phase 1b and randomised phase 2 trial. Lancet. 2016 Jul 30;388(10043):488-97.

4.    Lapalombella R, Sun Q, Williams K, et al: Selective inhibitors of nuclear export show that CRM1/XPO1 is a target in chronic lymphocytic leukemia. Blood 120:4621-4634, 2012

5.    Yoshimura M, Ishizawa J, Ruvolo V, et al: Induction of p53-mediated transcription and apoptosis by exportin-1 (XPO1) inhibition in mantle cell lymphoma. Cancer Sci 105:795-801, 2014

6.     Gounder MM, Zer A, Tap WD, Salah S, Dickson MA, Gupta AA, Keohan ML, Loong HH, D'Angelo SP, Baker S, Condy M, Nyquist-Schultz K, Tanner L, Erinjeri JP, Jasmine FH, Friedlander S, Carlson R, Unger TJ, Saint-Martin JR, Rashal T, Ellis J, Kauffman M, Shacham S, Schwartz GK, Abdul Razak AR. Phase IB Study of Selinexor, a First-in-Class Inhibitor of Nuclear Export, in Patients With Advanced Refractory Bone or Soft Tissue Sarcoma. J Clin Oncol. 2016 Sep 10;34(26):3166-74. doi: 10.1200/JCO.2016.67.6346. Epub 2016 Jul 25.

7.    Chi Y, Fang Z, Hong X, Yao Y, Sun P, Wang G, Du F, Sun Y, Wu Q, Qu G, Wang S, Song J, Yu J, Lu Y, Zhu X, Niu X, He Z, Wang J, Yu H, Cai J. Safety and Efficacy of Anlotinib, a Multikinase Angiogenesis Inhibitor, in Patients with Refractory Metastatic Soft-Tissue Sarcoma. Clin Cancer Res. 2018 Nov 1;24(21):5233-5238. doi: 10.1158/1078-0432.CCR-17-3766. Epub 2018 Jun 12.

8.     Mehta CR, Liu L, Theuer C. An Adaptive Population Enrichment Phase 3 Trial of TRC105 and Pazopanib Versus Pazopanib Alone in Patients with Advanced Angiosarcoma (TAPPAS Trial). Ann Oncol. 2018 Oct 24. doi: 10.1093/annonc/mdy464. [Epub ahead of print]

9.    Tap WD, Wainberg ZA, Anthony SP, Ibrahim PN, Zhang C, Healey JH, Chmielowski B, Staddon AP, Cohn AL, Shapiro GI, Keedy VL, Singh AS, Puzanov I, Kwak EL, Wagner AJ, Von Hoff DD, Weiss GJ, Ramanathan RK, Zhang J, Habets G, Zhang Y, Burton EA, Visor G, Sanftner L, Severson P, Nguyen H, Kim MJ, Marimuthu A, Tsang G, Shellooe R, Gee C, West BL, Hirth P, Nolop K, van de Rijn M, Hsu HH, Peterfy C, Lin PS, Tong-Starksen S, Bollag G. Structure-Guided Blockade of CSF1R Kinase in Tenosynovial Giant-Cell Tumor. N Engl J Med. 2015 Jul 30;373(5):428-37. doi: 10.1056/NEJMoa1411366.

10.   DuBois SG, Laetsch TW, Federman N, Turpin BK, Albert CM, Nagasubramanian R, Anderson ME, Davis JL, Qamoos HE, Reynolds ME, Cruickshank S, Cox MC, Hawkins DS, Mascarenhas L, Pappo AS. The use of neoadjuvant larotrectinib in the management of children with locally advanced TRK fusion sarcomas. Cancer. 2018 Nov 1;124(21):4241-4247.

11.   Fung HY, Chook YM: Atomic basis of CRM1-cargo recognition, release and inhibition. Semin Cancer Biol 27:52-61, 2014

12.  Tan DS, Bedard PL, Kuruvilla J, et al: Promising SINEs for embargoing nuclear-cytoplasmic export as an anticancer strategy. Cancer Discov 4:527-537, 2014

13.   Huang WY, Yue L, Qiu WS, et al: Prognostic value of CRM1 in pancreas cancer. Clin Invest Med 32:E315, 2009

14.   Noske A, Weichert W, Niesporek S, et al: Expression of the nuclear export protein chromosomal region maintenance/exportin 1/Xpo1 is a prognostic factor in human ovarian cancer. Cancer 112: 1733-1743, 2008

15.   Shen A, Wang Y, Zhao Y, et al: Expression of CRM1 in human gliomas and its significance in p27 expression and clinical prognosis. Neurosurgery 65: 153-159, 2009; discussion 159-160

16.     van der Watt PJ, Maske CP, Hendricks DT, et al: The Karyopherin proteins, Crm1 and Karyopherin beta1, are overexpressed in cervical cancer and are critical for cancer cell survival and proliferation. Int J Cancer 124:1829-1840, 2009

17.     Yao Y, Dong Y, Lin F, et al: The expression of CRM1 is associated with prognosis in human osteosarcoma. Oncol Rep 21:229-235, 2009

18.     Molena B, Sfriso P, Oliviero F, et al. Synovial colony-stimulating factor-1 mRNA expression in diffuse pigmented villonodular synovitis. Clin Exp Rheumatol 2011; 29: 547-50.

19.     Cupp JS, Miller MA, Montgomery KD, et al. Translocation and expression of CSF1 in pigmented villonodular synovitis, tenosynovial giant cell tumor, rheumatoid arthritis and other reactive synovitides. Am J Surg Pathol 2007; 31: 970-6.

20.     Duffy AG, Ma C, Ulahannan SV et al. Phase I and preliminary phase II study of TRC105 in combination with sorafenib in hepatocellular carcinoma. Clin Cancer Res 2017; 23(16): 4633–4641.

21.     Tian H, Huang JJ, Golzio C et al. Endoglin interacts with VEGFR2 to promote angiogenesis. Faseb J 2018.

22.     Sankhala K, Riedel R, Conry R et al. Every other week dosing of TRC105 (endoglin antibody) in combination with pazopanib in patients with advanced soft tissue sarcoma. In Connective Tissue Oncology Society Annual Meeting, November 2017, Maui, Hawaii.

23.     Skaper SD. The neurotrophin family of neurotrophic factors: an overview. Methods Mol Biol. 2012;846:1-12.

24.     Davis JL, Lockwood CM, Albert CM, Tsuchiya K, Hawkins DS, Rudzinski ER. Infantile NTRK-associated mesenchymal tumors. Pediatr Dev Pathol. 2018;21:68-78.

25.     Drilon A, Laetsch TW, Kummar S, et al. Efficacy of larotrectinib in TRK fusion–positive cancers in adults and children. N Engl J Med.2018;378:731-739.

Round  2

Reviewer 2 Report

The authors have greatly improved the flow and impact of the paper. It would be helpful to have the information about NY-ESO-1 given in the author's response also in the paper. It helps to clarify why NY-ESO-1 might be a potential target. There are a few minor grammatical errors that need to be corrected. 

ex. Line 187: Trabectedin is a tetrahydroisoquinoline alkaloid that was initially isolated from the marine tunicate Ecteinascidia turbinate. 

Author Response

Dear reviewer, we sincerely thank you for your considering our manuscript entitled “Current therapeutic targets for bone and soft-tissue sarcomas” for publication in International Journal of Molecular Sciences. We are very grateful for your prompt attention and thorough review. Based on the reviewers’ comments, we have revised the manuscript and have addressed all of the concerns brought up. Please see below for our point-by point responses to each of the reviewers’ comments. If there are any further issues that need to be resolved to improve our manuscript, please let us know.

Newly added or changed parts are indicated in red in the revised manuscript.

Comments and Suggestions for Authors 2

The authors have greatly improved the flow and impact of the paper. It would be helpful to have the information about NY-ESO-1 given in the author's response also in the paper. It helps to clarify why NY-ESO-1 might be a potential target. There are a few minor grammatical errors that need to be corrected.

ex. Line 187: Trabectedin is a tetrahydroisoquinoline alkaloid that was initially isolated from the marine tunicate Ecteinascidia turbinate.

Response: Thank you for your suggestion. Following sentences were added to the section.

Cancer testis antigens have emerged as both a diagnostic marker and a therapeutic target in malignant tumors. Cancer testis antigen NY-ESO-1 (New York esophageal squamous cell carcinoma 1) is encoded by the CTAG1B gene, and expressed in 10–50% of metastatic melanomas, breast, prostate, thyroid, and ovarian cancers [6-9].

The line 187 was corrected according to the reviewer’s comment.

References:

1.  Chen YT, Scanlan MJ, Sahin U, et al: A testicular antigen aberrantly expressed in human cancers detected by autologous antibody screening. Proc Natl Acad Sci U S A 94:1914-1918, 1997

2.   Barrow C, Browning J, MacGregor D, et al: Tumor antigen expression in melanoma varies according to antigen and stage. Clin Cancer Res 12: 764-771, 2006

3.   Goydos JS, Patel M, Shih W: NY-ESO-1 and CTp11 expression may correlate with stage of progression in melanoma. J Surg Res 98:76-80, 2001

4.   Gure AO, Chua R, Williamson B, et al: Cancertestis genes are coordinately expressed and are markers of poor outcome in non-small cell lung cancer. Clin Cancer Res 11:8055-8062, 2005

Reviewer 3 Report

This revised manuscript by Miwa et al reviews current therapies for bone and soft tissue sarcomas. The revised version is substantially more comprehensive, and is a clear improvement from the initial version.

Current issues include:

The authors should be careful not to overstate conclusions, and should be more precise in stating the benefits of particular therapies. For example:

- in line 15, the sentence should be modified to say that recent basic science research has revealed some of the mechanisms of progression and metastasis…

- in line 28, the authors say that chemotherapy significantly improves the outcomes of sarcomas, but only surgery can offer cure. The authors should specify what outcomes are improved.

- in line 183, eribulin is described as “one of the best treatment options in advanced sarcoma”, but mention should be made in its prioritization compared to doxorubicin.

- in line 198, do the results of the TRUST tiral really suggest that “trabectidin might improve survival in patients with advanced soft tissue sarcoma”? This conclusion is not necessarily supported by the information presented.

2.       Formatting issues in Table 1 should be corrected. The heading “serious adverse effects” should be replaced by “grade 3-4 toxicities.” Would consider adding olaratumab to this table, as it is now FDA approved for treatment of soft tissue sarcoma.

3.       It would be helpful to compare and contrast the toxicity and activity of sorafenib vs pazopanib. Also, mention should be made of the combination of sorafenib and everolimus for osteosarcoma (Lancet Oncol 2015;16:98-107).

4.       The authors should comment on the possible target(s) of imatinib in sarcomas (e.g., probably not BCR-ABL) in both the manuscript and Table 1.

5.       Consideration could be given to grouping together the discussion of targeted agents, both established and new, rather than separating these discussions as is currently done. A table summarizing the agents discussed in Section 2.6 would be helpful.

6.       Mention should be made of the ANNOUNCE phase III study to better define the benefits of adding olaratumab to doxorubicin.

7.       The use of an English scientific editor may further help with the readability of the manuscript. In addition, the word “promising” is used repeatedly throughout the manuscript, and perhaps more varied descriptions of the different regimens could be considered.

Author Response

Dear reviewer, we sincerely thank you for your considering our manuscript entitled “Current therapeutic targets for bone and soft-tissue sarcomas” for publication in International Journal of Molecular Sciences. We are very grateful for your prompt attention and thorough review. Based on the reviewers’ comments, we have revised the manuscript and have addressed all of the concerns brought up. Please see below for our point-by point responses to each of the reviewers’ comments. If there are any further issues that need to be resolved to improve our manuscript, please let us know.

Newly added or changed parts are indicated in red in the revised manuscript.

This revised manuscript by Miwa et al reviews current therapies for bone and soft tissue sarcomas. The revised version is substantially more comprehensive, and is a clear improvement from the initial version.

Current issues include:

The authors should be careful not to overstate conclusions, and should be more precise in stating the benefits of particular therapies. For example:

- in line 15, the sentence should be modified to say that recent basic science research has revealed some of the mechanisms of progression and metastasis…

Response: Thank you for your recommendation. The sentence was modified to “basic science research has revealed some of the mechanisms of progression and metastasis…”. 

- in line 28, the authors say that chemotherapy significantly improves the outcomes of sarcomas, but only surgery can offer cure. The authors should specify what outcomes are improved.

Response: Thank you for your suggestion. The following sentences were added to the manuscript.

Before introduction of chemotherapy, long term survival was observed in only 20–40% of patients with bone sarcomas and in only 35% of patients with soft tissue sarcomas [Presant CA, 1981]. Since 1970s, chemotherapy significantly improved the outcomes of sarcoma, and 5-year survival of 60–80% have been reported for patients receiving chemotherapy and with surgical resection[Bielack SS, J Clin Oncol, 2002; Meyers PA, J Clin Oncol, 2005; Goorin AM, J Clin Oncol, 2003; Bacci G, Cancer, 2003; Crews KR, Cancer, 2004].

References:

1.  Presant CA, Lowenbraun S, Bartolucci AA, Smalley RV. Metastatic sarcomas: chemotherapy with adriamycin, cyclophosphamide, and methotrexate alternating with actinomycin D, DTIC, and vincristine. Cancer. 47(3), 457–465 (1981).

2.   Bielack SS, Kempf-Bielack B, Delling G, Exner GU, Flege S, Helmke K, Kotz R, Salzer-Kuntschik M, Werner M, Winkelmann W, Zoubek A, Jürgens H, Winkler K. Prognostic factors in high-grade osteosarcoma of the extremities or trunk: an analysis of 1,702 patients treated on neoadjuvant cooperative osteosarcoma study group protocols. J Clin Oncol. 20(3), 776–790 (2002).

3.   Meyers PA, Schwartz CL, Krailo M, Kleinerman ES, Betcher D, Bernstein ML, Conrad E, Ferguson W, Gebhardt M, Goorin AM, Harris MB, Healey J, Huvos A, Link M, Montebello J, Nadel H, Nieder M, Sato J, Siegal G, Weiner M, Wells R, Wold L, Womer R, Grier H. Osteosarcoma: a randomized, prospective trial of the addition of ifosfamide and/or muramyl tripeptide to cisplatin, doxorubicin, and high-dose methotrexate. J Clin Oncol. 23(9), 2004–2011 (2005).

4.  Goorin AM, Schwartzentruber DJ, Devidas M, Gebhardt MC, Ayala AG, Harris MB, Helman LJ, Grier HE, Link MP; Pediatric Oncology Group. Presurgical chemotherapy compared with immediate surgery and adjuvant chemotherapy for nonmetastatic osteosarcoma: Pediatric Oncology Group Study POG-8651. J Clin Oncol. 21(8), 1574–1580 (2003).

5.   Bacci G, Bertoni F, Longhi A, Ferrari S, Forni C, Biagini R, Bacchini P, Donati D, Manfrini M, Bernini G, Lari S. Neoadjuvant chemotherapy for high-grade central osteosarcoma of the extremity. Histologic response to preoperative chemotherapy correlates with histologic subtype of the tumor. Cancer. 97(12), 3068–3075 (2003).

6.   Crews KR, Liu T, Rodriguez-Galindo C, Tan M, Meyer WH, Panetta JC, Link MP, Daw NC. High-dose methotrexate pharmacokinetics and outcome of children and young adults with osteosarcoma. Cancer. 100(8), 1724–1733 (2004).

- in line 183, eribulin is described as “one of the best treatment options in advanced sarcoma”, but mention should be made in its prioritization compared to doxorubicin.

Response: Thank you for pointing this out. Because the expression “one of the best treatment options” was overstatement, the “best” was deleted from the manuscript.

- in line 198, do the results of the TRUST tiral really suggest that “trabectidin might improve survival in patients with advanced soft tissue sarcoma”? This conclusion is not necessarily supported by the information presented.

Response: Thank you for pointing this out. The conclusion was deleted from the section.

2.   Formatting issues in Table 1 should be corrected. The heading “serious adverse effects” should be replaced by “grade 3-4 toxicities.” Would consider adding olaratumab to this table, as it is now FDA approved for treatment of soft tissue sarcoma.

Response: Thank you for your suggestion. The heading “serious adverse effects” was replaced by “grade 3-4 toxicities.” Information of olaratumab was added to Table 1.

3.   It would be helpful to compare and contrast the toxicity and activity of sorafenib vs pazopanib. Also, mention should be made of the combination of sorafenib and everolimus for osteosarcoma (Lancet Oncol 2015;16:98-107).

Response: Thank you for your suggestion.

We agree that it is useful to compare the toxicity and activity of sorafenib vs pazopanib. However, we could not find the report comparing the toxicity and activity of sorafenib with pazopanib. Following sentence was added to the manuscript.

A phase II study with the combination of sorafenib and everolimus in patients with unresectable osteosarcoma was reported in 2015 by Italian Sarcoma Group [Grignani G, 2015]. In the study, 38 patients received 800 mg sorafenib plus 5 mg everolimus once a day until disease progression or unacceptable toxic effects. The study showed that 17 of 38 patients (45%) were progression free at 6 months.

Reference:

Grignani G, Palmerini E, Ferraresi V, D'Ambrosio L, Bertulli R, Asaftei SD, Tamburini A, Pignochino Y, Sangiolo D, Marchesi E, Capozzi F, Biagini R, Gambarotti M, Fagioli F, Casali PG, Picci P, Ferrari S, Aglietta M; Italian Sarcoma Group. Sorafenib and everolimus for patients with unresectable high-grade osteosarcoma progressing after standard treatment: a non-randomised phase 2 clinical trial. Lancet Oncol. 2015 Jan;16(1):98-107.

4.  The authors should comment on the possible target(s) of imatinib in sarcomas (e.g., probably not BCR-ABL) in both the manuscript and Table 1.

Response: Thank you for your pointing this out. The possible target of imatinib was corrected as follows.

Imatinib mesylate, a tyrosine kinase inhibitor originally developed as an inhibitor of the BCR-ABL tyrosine kinase, was also found to inhibit PDGFR and c-kit.

5.   Consideration could be given to grouping together the discussion of targeted agents, both established and new, rather than separating these discussions as is currently done. A table summarizing the agents discussed in Section 2.6 would be helpful.

Response: Thank you for your suggestion. The targeted therapies were combined and discussed in one section.

6.   Mention should be made of the ANNOUNCE phase III study to better define the benefits of adding olaratumab to doxorubicin.

Response: Thank you for your suggestion. The following sentence was added. Phase III study with olaratumab and doxorubicin in patients with advanced sarcomas is ongoing (NCT02451943). This study may determine the benefits of adding olaratumab to doxorubicin.

7.   The use of an English scientific editor may further help with the readability of the manuscript. In addition, the word “promising” is used repeatedly throughout the manuscript, and perhaps more varied descriptions of the different regimens could be considered.

Response: Thank you for your suggestion. The word “promising” was replaced by other expressions in the manuscript.